# Do LLMs Understand Visual Anomalies?
# Uncovering LLM's Capabilities in Zero-shot Anomaly Detection

## ABSTRACT

Large vision-language models (LVLMs) are markedly proficient in deriving visual representations guided by natural language. Recent explorations have utilized LVLMs to tackle zero-shot visual anomaly detection (VAD) challenges by pairing images with textual descriptions indicative of normal and abnormal conditions, referred to as *anomaly prompts*. However, existing approaches depend on static anomaly prompts that are prone to cross-semantic ambiguity, and prioritize global image-level representations over crucial local pixel-level image-to-text alignment that is necessary for accurate anomaly localization. In this paper, we present ALFA, a training-free approach designed to address these challenges via a unified model. We propose a run-time prompt adaptation strategy, which first generates informative anomaly prompts to leverage the capabilities of a large language model (LLM). This strategy is enhanced by a contextual scoring mechanism for per-image anomaly prompt adaptation and cross-semantic ambiguity mitigation. We further introduce a novel fine-grained aligner to fuse local pixel-level semantics for precise anomaly localization, by projecting the image-text alignment from global to local semantic spaces. Extensive evaluations on the challenging MVTec and VisA datasets confirm ALFA's effectiveness in harnessing the language potential for zero-shot VAD, achieving significant PRO improvements of 12.1% on MVTec AD and 8.9% on VisA compared to state-of-the-art zero-shot VAD approaches.

## CCS CONCEPTS

• **Computing methodologies → Computer vision tasks**.

## KEYWORDS

Visual Anomaly Detection, Large Vision-language Model, Zero-shot, Industrial Manufacturing

**ACM Reference Format:**
Anonymous Author(s). 2024. Do LLMs Understand Visual Anomalies? Uncovering LLM's Capabilities in Zero-shot Anomaly Detection. In *Proceedings of Make sure to enter the correct conference title from your rights confirmation email (MM '24)*. ACM, New York, NY, USA, 10 pages. https://doi.org/XXXXXXX.XXXXXXX

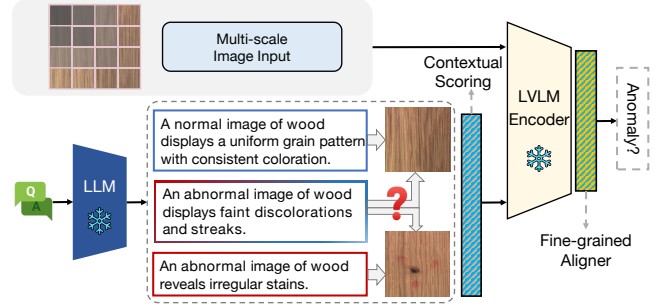

**Figure 1: Overview of ALFA, a training-free zero-shot VAD model focusing on vision-language synergy. The first and third prompts are generated by an LLM to describe normal and abnormal images, respectively. The second prompt, however, shows an ambiguous description, posing a challenge in accurately determining the image label, a phenomenon known as cross-semantic ambiguity.**

## 1 INTRODUCTION

Visual anomaly detection (VAD) has gained momentum in a wide spectrum of domains, including industrial quality control [2, 37], video surveillance [14, 44], medical diagnostics [10, 33, 53] and etc. This complex task involves both anomaly classification and localization for images, i.e., image-level and pixel-level anomaly detection. Inevitably, VAD faces two fundamental challenges due to the nature of its detection targets. First, the diversity of image objects makes the categories of anomalies a long-tail distribution [39]. To address the diverse range of images, a universal, category-agnostic model is required, as opposed to the traditional approach of deploying dedicated models for specific visual inspection tasks. The latter approach is unscalable and inefficient due to the long tail characteristic of the problem [26, 31]. Second, anomaly images are rare and have great variations [6, 19, 60]. In real-world applications like industrial VAD, collecting a sufficient and diverse training sample set is both costly and time-consuming. This scarcity complicates the training of traditional one-class or unsupervised VAD models, especially in cold-start scenarios [37].

The introduction of zero-shot methods offers a promising solution to these challenges. The emergence of large-scale models [23, 35] has revolutionized VAD profoundly. Recently, several large vision-language models (LVLMs) have been introduced for zero-shot VAD [4, 13, 17, 19, 58]. These works harness the exceptional generalization ability of LVLMs, pre-trained on millions of image-text pairs, which showcase promising zero-shot performance in both seen and unseen objects. Nonetheless, due to the inherent lack of comprehensive information on data and the absence of explicit supervision, the zero-shot regime remains particularly challenging, with significant potential yet to be exploited compared to fully-supervised benchmarks.

There are two major limitations. First, existing works rely on fixed textual descriptions of images, termed *anomaly prompts*, including both abnormal and normal prompts. In LVLM-based VAD, anomaly prompts elucidate the semantics of normalities and anomalies and guide the vision modules on how the two states are defined, the quality of which, therefore, plays a critical role in the zero-shot detection capability of LVLMs. The current practice of manually crafting prompts demands extensive domain expertise and considerable time, while also facing the challenge of cross-semantic ambiguity, which is illustrated in Figure 1 and will be discussed in depth in Sec. 4.2. This limitation calls for more informative and adaptive anomaly prompts. Second, although LVLMs, trained for image-text cross-modal alignment, can detect anomalies globally by aligning image-level representations with anomaly prompts, they face difficulties in localizing anomalies precisely, i.e., achieving pixel-level detection. Such local pixel-level alignment is central to zero-shot anomaly segmentation [19].

In this paper, we focus on zero-shot modeling and address the limitations of existing models with a proposal called ALFA – **A**daptive **L**LM-empowered model for zero-shot visual anomaly detection with **F**ine-grained **A**lignment. We introduce a run-time prompt adaptation strategy to efficiently generate informative and adaptive anomaly prompts, which obviates the need for laborious expert creation and tackles cross-semantic ambiguity. Leveraging the zero-shot capabilities of an LLM [3] that is renowned for its proficient instruction-following abilities [24], this strategy automatically generates a diverse range of informative anomaly prompts for VAD. Next, we present a contextual scoring mechanism to adaptively tailor a set of anomaly prompts for each query image. To fully excavate the local pixel-level semantics, we further propose a novel fine-grained aligner that generalizes the image-text alignment projection from global to local semantic space for precise anomaly localization. This cross-modal aligner enables ALFA to achieve global and local VAD within one unified model without requiring additional data or fine-tuning. We summarize our main contributions as follows:

- We identify a previously unaddressed issue of cross-semantic ambiguity. In response, we present ALFA, an adaptive LLM-empowered model for zero-shot VAD, effectively resolving this challenge without the need for extra data or fine-tuning.
- We propose a run-time prompt adaptation strategy that effectively generates informative anomaly prompts and dynamically adapts a set of anomaly prompts on a per-image basis.
- We develop a fine-grained aligner that learns global to local semantic space projection, and then, generalizes this projection to support precise pixel-level anomaly localization.
- Our comprehensive experiments validate ALFA's capacity for zero-shot VAD across diverse datasets. Moreover, ALFA can be readily extended to the few-shot setting, which achieves state-of-the-art results that are on par or even outperform those of full-shot and fine-tuning-based methods.

## 2 RELATED WORK

**Vision-language modeling.** Large Language Models (LLMs) such as GPT [3] and LLaMA [46] have achieved remarkable performance on NLP tasks. Since the introduction of CLIP [35], large Visual-Language Models (LVLMs) like MiniGPT-4 [59], BLIP-2 [27], and PandaGPT [43] have shown promise across a range of language-guided tasks. Without additional fine-tuning, text prompts can be used to extract knowledge in the downstream image-related tasks such as zero-shot classification [32], object detection [21], and segmentation [49]. Consequently, LVLMs offer the potential to advance language-guided anomaly detection in a zero-shot manner. In this paper, we delve deeper into exploring how to optimize the utilization of LVLMs for visual anomaly detection (VAD).

**Visual anomaly detection.** Given the scarcity of anomalies, conventional VAD approaches primarily focus on unsupervised or self-supervised methods relying exclusively on normal images. These approaches fall into two main categories: *generative models* [26, 31, 36, 50, 56] that utilize an encoder-decoder framework to minimize the reconstruction error, and *feature embedding-based models* that detect anomalies by discerning variations in feature distribution between normal and abnormal images. The latter includes one-class methods [30, 38, 47], memory-based models [16, 34, 37] and knowledge distillation models [1, 12, 40, 52] hinging on the knowledge captured by networks pre-trained on large dataset.

Recent research has delved into zero-shot VAD, reducing reliance on either normal or abnormal images and offering a unified anomaly detection model applicable across various image categories [4, 13, 17, 19, 28, 58, 61]. Notably, WinCLIP [19] pioneers the potential of language-driven zero-shot VAD, leveraging CLIP to extract and aggregate multi-scale image features. MuSc [28] proposes a looser zero-shot approach that utilizes a pre-trained Vision Transformer (ViT) [15] to extract patch-level features and assesses anomaly scores by comparing the similarity of patches between the query image and hundreds of unlabeled images. However, these approaches still suffer from several limitations, which require manual prompt crafting, intricate post-processing of extra data, and additional fine-tuning. In contrast, ALFA is a training-free model for zero-shot VAD, obviating the need for extra data or additional fine-tuning, and generates informative and adaptive prompts without costly manual design.

**Probing through visual prompt engineering.** In VAD, prompts describe image content to assess anomaly levels by aligning with both normal and abnormal prompts. Traditional prompt engineering [25, 29, 42] that adjusts the model with learnable tokens is unsuitable due to data requirements. Existing efforts [1, 4, 8, 13, 17, 19, 45] typically hand craft numerous descriptions for detection, e.g., WinCLIP [19] using a compositional prompt ensemble and SAA [4] employing a prompt regularization strategy. However, these predefined-based approaches are inefficient and suboptimal. Recent studies have explored using LLMs to generate prompts for object recognition [21, 51], potentially alleviating the challenge of inefficient prompt design. However, directly applying this approach to VAD tasks leads to cross-semantic ambiguity, caused by textual descriptions encompassing various aspects of an image, some of which may not be present or prominent in the query image. To avoid this, this paper proposes a run-time prompt adaptation strategy utilizing an LLM, coupled with a contextual scoring mechanism, to generate informative and adaptive prompts, which effectively addresses the cross-semantic issue.

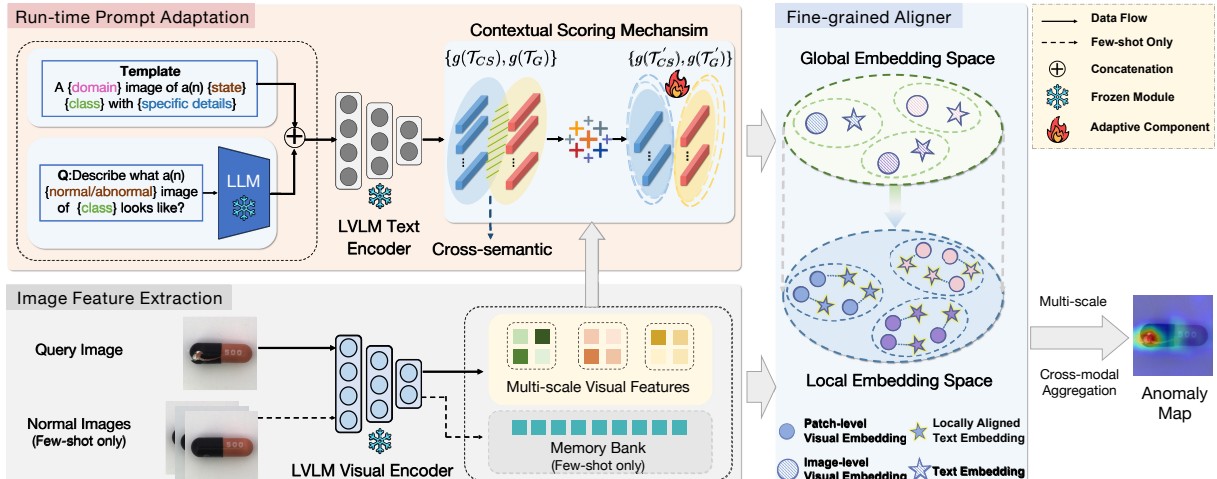

**Figure 2: Workflow of ALFA with the run-time prompt adaptation strategy, which generates informative prompts and adaptively manages a collection of prompts on a per-image basis via a contextual scoring mechanism. Furthermore, a fine-grained aligner is introduced to generalize the alignment projection from global to local for precise anomaly localization.**

## 3 PRELIMINARY

### 3.1 Visual Anomaly Detection

*Anomaly detection* aims to detect data samples that deviate from the majority or exhibit unusual patterns. Particularly, this paper focuses on *visual anomaly detection* (VAD), the objectives of which are to (1) detect anomalies globally for images, and (2) localize anomalies for pixels of each image locally, as formulated below:

DEFINITION 1 (VISUAL ANOMALY DETECTION). *Given an image* $x \in \mathbb{R}^{H \times W \times C}$, *VAD aims to predict whether* $x$ *and all its individual pixel* $x_{ij}$, *are anomalous or not, where* $0 \leq i < H$ *and* $0 \leq j < W$.

In this study, we seek to develop a category-agnostic VAD approach that exhibits generalizability across categories $c_i \in C$, allowing the model to readily adapt to new categories without model retraining or parameter fine-tuning. Formally, for $\forall x \in c_i$, VAD can be achieved by computing anomaly scores $S_i, S_p = \mathcal{M}(x; \Theta_m)$ for both global image-level $S_i \in [0, 1]$ and local pixel-level $S_p \in [0, 1]^{H \times W \times 1}$, using a detection model $\mathcal{M}(\cdot)$ parameterized by $\Theta_m$.

### 3.2 Zero-shot Anomaly Detection with LVLMs

LVLMs provide a unified representation for both vision and language modalities, leveraging contrastive learning-based [7] pre-training approaches to learn a shared embedding space. Given million-scale image-text pairs $\{(x_j^{c_i}, t_j^{c_i}) | 0 \leq j < n_i, c_i \in C\}$, where $n_i$ is the number of pairs in category $c_i$, LVLMs train an image encoder $f(\cdot)$ and a text encoder $g(\cdot)$ by maximizing the correlation between $f(x_j^{c_i})$ and $g(t_j^{c_i})$ measured in cosine similarity $<f(x_j^{c_i}), g(t_j^{c_i})>$. This strategy effectively aligns images with text prompts in LVLMs.

LVLMs can be adopted for zero-shot language-guided anomaly detection for images. For instance, given an image $x_j$, two predefined text templates, *i.e.,* "a photo of a normal $[c_i]$" and "a photo of an abnormal $[c_i]$" and the extracted text tokens $t^+$ and $t^-$ correspondingly, anomaly detection is achieved by exploiting the visual

information extracted by the image encoder and computing an *anomaly score* for category $c_i$:

$$S(x_j^{c_i}) = \frac{\exp(<f(x_j^{c_i}), g(t^-)>)}{\sum_{t \in \{t^+, t^-\}} \exp(<f(x_j^{c_i}), g(t)>)} \quad (1)$$

which basically measures the proximity of the image $x_j$ to the abnormal text template of category $c_i$ by tapping into the vision-language alignment capability of LVLMs.

## 4 METHODOLOGY

### 4.1 Overview

In this paper, we propose an LLM-empowered LVLM model ALFA for zero-shot VAD. As shown in Figure 2, ALFA first introduces a run-time prompt (RTP) adaptation strategy to generate informative prompts and adaptively manage a collection of prompts on a per-image basis via a contextual scoring mechanism (see Sec. 4.2). Unlike conventional run-time adaptation approaches, which require fine-tuning their pre-trained models during inference, our strategy functions without the requirement for any parameter update. Furthermore, we present a training-free fine-grained aligner to bridge the cross-modal gap between global and local semantic spaces, enabling precise zero-shot anomaly localization (see Sec. 4.3).

### 4.2 Run-time Prompt Adaptation

The quality of textual prompts significantly influences the zero-shot detection capabilities of LVLMs. Figure 3 provides a visual overview of our prompt generation and adaptation process, with more details elaborated below.

**Informative prompt generation.** Staying in line with the prompt learning trend [22, 57], we first employ general expert knowledge to initialize the contrastive-state prompts, unlocking LVLMs' knowledge guided by language. We design unified templates with specific contents to generate comprehensive prompts covering task-relevant concepts thoroughly, which contrasts with prior approaches that

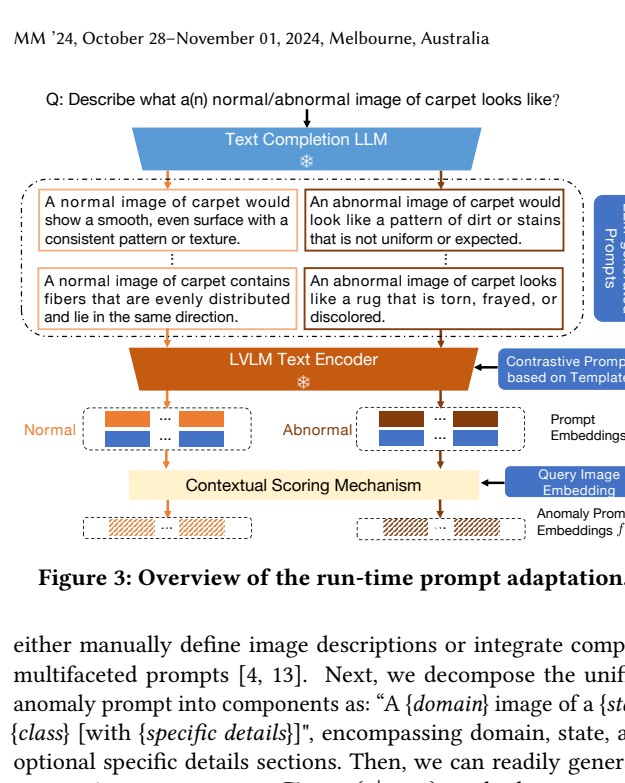

Figure 3: Overview of the run-time prompt adaptation.

either manually define image descriptions or integrate complex multifaceted prompts [4, 13]. Next, we decompose the unified anomaly prompt into components as: "A {*domain*} image of a {*state*} {*class*} [with {*specific details*}]", encompassing domain, state, and optional specific details sections. Then, we can readily generate contrastive-state prompts $\mathcal{T}_{CS} = \{\mathbf{t}_{cs}^+, \mathbf{t}_{cs}^-\}$ as the base anomaly detector, where $\mathbf{t}_{cs}^+ = \{t_{cs,0}^+, \cdots, t_{cs,n_{cs}^+}^+\}, \mathbf{t}_{cs}^- = \{t_{cs,0}^-, \cdots, t_{cs,n_{cs}^-}^-\}$. $n_{cs}^+$ and $n_{cs}^-$ indicate the number of normal and abnormal prompts generated by the unified template.

Recognizing the potential for domain gaps to introduce language ambiguity, especially with a generic prompt, base anomaly detector derived from the unified template falls short. LLMs are repositories of extensive world knowledge spanning diverse domains, serving as implicit knowledge bases that facilitate effortless natural language queries [9]. This knowledge includes visual descriptors, enabling LLMs to furnish insights into image features. To avoid the costly and non-scalable practice of manually crafting prompts using domain-specific knowledge, we efficiently tap into LLMs for more informative prompts. To this end, we design prompts to query an LLM, e.g., "How to identify an abnormal bottle in an image?". Using this approach, we can derive precise descriptions of a wide range of objects in normal and abnormal states as $\mathcal{T}_G = \{\mathbf{t}_g^+, \mathbf{t}_g^-\}$, where $\mathbf{t}_g^+ = \{t_{g,0}^+, \cdots, t_{g,n_g^+}^+\}, \mathbf{t}_g^- = \{t_{g,0}^-, \cdots, t_{g,n_g^-}^-\}$, and $n_g^+$ and $n_g^-$ indicate the number of normal and abnormal prompts generated by the LLM.

**Remark.** In dealing with the diverse and unpredictable nature of anomalies, language offers the essential information to discern defects from acceptable deviations. Building upon the insights from [32], we can enhance interpretability in VAD decisions by leveraging the capabilities of LLMs. Specifically, LLM can be employed to produce feature descriptions regarding anomalies. These descriptions can be provided to the LVLM to compute the logarithmic probability of each description pertaining to the image query. By examining the descriptors with high scores, we can gain insights into the model's decision. More details are provided in Section 5.4.

**Cross-semantic ambiguity.** In an ideal scenario, LVLMs for zero-shot VAD should be capable of recognizing the close correlation between normal images and their respective normal prompts, while

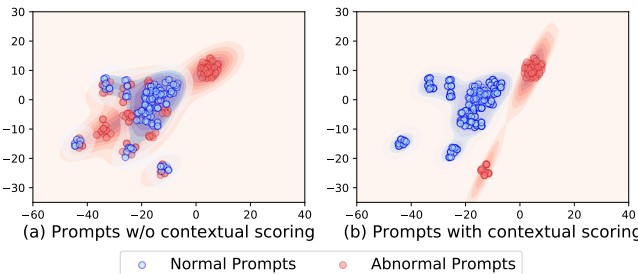

Figure 4: Visualization of ALFA's semantic space.

identifying a more distant association with abnormal prompts. The relative distances to normal and abnormal prompts are crucial for LVLMs to detect anomalies effectively. However, by visualizing the semantic space of LVLMs, we observed an overlap and intersection in the feature distributions of both normal and abnormal prompts, as depicted in Figure 4 (a). This leads to situations where the features of certain anomalous images are closer to normal prompts while being distant from certain abnormal prompts.

We refer to this phenomenon as `cross-semantic ambiguity`. We attribute this phenomenon to the intricate nature of textual descriptions and the semantic correlation between text and image. This is exacerbated by prompts covering diverse aspects of the image, some of which might not be salient or even absent in certain images. Anomaly detection is thus susceptible to cross-semantic ambiguity poisoning. Therefore, there is a pressing need for an effective remedy to adaptively manage a set of normal and abnormal anomaly prompts corresponding to each query image without semantic overlap.

**Contextual scoring mechanism.** To address the persistent challenge of the cross-semantic ambiguity in VAD, we propose a contextual scoring mechanism, which adaptively adjusts a set of anomaly prompts on a per-image basis.

Specifically, given a query image $x \in \mathbb{R}^{H \times W \times C}$ and the vanilla anomaly prompts $\mathcal{T}_{vanilla} := \{\mathcal{T}_{CS}, \mathcal{T}_G\} = \{\mathbf{t}_{cs}^+, \mathbf{t}_{cs}^-, \mathbf{t}_g^+, \mathbf{t}_g^-\}$, their embeddings can be obtained using the pre-trained image and text encoders of LVLMs, denoted as $f(x) \in \mathbb{R}^d$ and $g(t) \in \mathbb{R}^d$, where $t \in \mathcal{T}_{vanilla}$ and $d$ denotes the dimension of the latent semantic space. We calculate the cosine similarity between the embeddings of the query image $x$ and normal $\{\mathbf{t}_{cs}^+, \mathbf{t}_g^+\}$ and abnormal prompts $\{\mathbf{t}_{cs}^-, \mathbf{t}_g^-\}$ respectively as:

$$d_i^+(x) = \ <f(x), g(t_i^+)>, t_i^+ \in \{\mathbf{t}_{cs}^+, \mathbf{t}_g^+\} \quad (2)$$

$$d_j^-(x) = \ <f(x), g(t_j^-)>, t_j^- \in \{\mathbf{t}_{cs}^-, \mathbf{t}_g^-\} \quad (3)$$

Ideally, the distances between images and prompts for normal and abnormal categories should fall into two non-overlapping intervals. Specifically, normal images should be closer to normal prompts, while their distance to abnormal prompts should be farther, and vice versa for abnormal images. However, in practice, considering the heterogeneous nature of textual descriptions, not all descriptions of normal or abnormal conditions can be observed in a single image, which leads to the presence of some redundant or even noisy prompts that could negatively impact the model's detection performance. In this regard, we formulate the contextual score as a logistic function [20] to quantify the prompt's impact in discerning abnormalities from normal occurrences. For each

prompt $t \in \mathcal{T}_{vanilla}$, its contextual score is calculated as follows,

$$\mathcal{S}_c(t) = \frac{1}{1 + e^{-k \cdot \left\| \mathcal{D}(d_x(t), d_i^+) - \mathcal{D}(d_x(t), d_j^-) \right\|}} \qquad (4)$$

where $d_x(t)$ represents the cosine similarity between the prompt $t$ and the query image $x$, and $k$ is an adjustment parameter that controls the slope of the scoring function. Empirically, we set $k$ to 1, ensuring the scoring function exhibits a moderate rate of change beyond the interval. $\mathcal{D}(\cdot, \cdot)$ is used to calculate the distance between a point and an interval as follows,

$$\mathcal{D}(d_x(t), d_i^+) = \max(0, \max(\min_i\{d_i^+(x)\} - d_x(t), d_x(t) - \max_i\{d_i^+(x)\}))$$

$$\mathcal{D}(d_x(t), d_j^-) = \max(0, \max(\min_j\{d_j^-(x)\} - d_x(t), d_x(t) - \max_j\{d_j^-(x)\}))$$

The contextual score $\mathcal{S}_c(t)$ of the prompt $t$ is constrained within the range $[0, 1]$. Considering the interval $[\min_i\{d_i^+(x)\}, \max_i\{d_i^+(x)\}]$ and $[\min_i\{d_i^-(x)\}, \max_i\{d_i^-(x)\}]$, when the distance between the prompt and the query image in the semantic space $d_x(t)$ places farther from another interval than the one it belongs to, the contextual score approaches 1, indicating a strong relevant, and vice versa. In cases where the distance $d_x(t)$ straddles both intervals, the score settles at 0, indicating an indeterminate relevance. Therefore, during inference, we employ the contextual scoring mechanism to filter out prompts with a contextual score of 0, retaining only those in non-overlapping intervals, represented as $\mathcal{T} := \{\mathcal{T}^+, \mathcal{T}^-\}$, with $\mathcal{T}^+$ and $\mathcal{T}^-$ representing the normal and abnormal prompts, respectively.

We outline the procedure of RTP adaptation in Algorithm 1, and visualize the feature distribution of prompts processed through the contextual scoring mechanism in Figure 4 (b), which demonstrates that the proposed contextual score effectively addressed the cross-semantic ambiguity. Notably, the anomaly prompt $\mathcal{T}$ varies depending on the specific query image, which aligns with the intuitive notion that prompts and their numbers tailored to different object classes should naturally differ. Even within the same class, different images necessitate different emphases on individual prompts. Consequently, the implementation of the contextual scoring mechanism offers an adaptive approach to managing a set of prompts on a per-image basis, which enables the selected prompts that are better suited to the unique characteristics of each query image, thus enhancing the overall effectiveness of anomaly detection.

## 4.3 Fine-grained Aligner

Since anomaly localization requires predicting anomalies at the pixel-level, acquiring dense visual features is necessary. However, LVLMs enforce cross-modal alignment globally for images and text, creating a cross-modal gap between the global prompt embeddings and local patch token embeddings. WinCLIP [19] attempts to address this issue by employing a sliding window to generate patch embeddings in a manner that simulates processing the global image instead of using patch-wise embeddings from the penultimate feature map. However, the localized patch may not encompass the description of the global image in the text prompt, leading to suboptimal performance. While AnomalyGPT [17] achieves alignment by generating pseudo-anomaly samples and introducing additional training, which is operationally intricate and lacks efficiency. Consequently, we propose a training-free fine-grained aligner to explicitly model the mapping between global and local semantic spaces.

---

**Algorithm 1** Run-time Prompt Adaptation

---

**Input:** Query image $x$, pre-trained image encoder $f(\cdot)$, pre-trained text encoder $g(\cdot)$
**Output:** Anomaly prompts $\mathcal{T} := \{\mathcal{T}^+, \mathcal{T}^-\}$
**Initialization:** Template-based prompt generator $\mathcal{G}_T$, LLM-based prompt generator $\mathcal{G}_L$

1: Generate $\mathcal{T}_{CS}$ by the template-based prompt generator $\mathcal{G}_T$
2: Generate $\mathcal{T}_G$ by the LLM-based prompt generator $\mathcal{G}_L$
3: Caculate the the cosine similarity between $f(x)$ and $g(t)$ as Eq.(2) and Eq.(3), $t \in \mathcal{T}_{vanilla} = \{\mathcal{T}_{CS}, \mathcal{T}_G\}$
4: **for** $t$ in $\mathcal{T}_{vanilla}$ **do**
5:  Calculate the contextual score $\mathcal{S}_c(t)$ using Eq.(4)
6:  **if** $\mathcal{S}_c(t) > 0$ **then**
7:   Add $t$ into $\mathcal{T}$
8:  **end if**
9: **end for**
10: **return** Anomaly prompts $\mathcal{T}$

---

Given a query image $x$ and its corresponding anomaly prompts $\mathcal{T}$, their embeddings can be denoted as $f(x) \in \mathbb{R}^d$ and $g(t) \in \mathbb{R}^d$, where $t \in \mathcal{T}$ and $d$ denotes the dimension of the latent space. Mathematically, the encoder architecture consists of vision transformers (ViT) [15] based on multi-head self-attention (MHSA) and a feed-forward network (FFN) with layer normalization (LN) and residual connections that can be expressed as:

$$\hat{z}^l = \text{MHSA}(\text{LN}(z^{l-1})) + z^{l-1} \qquad (5)$$

$$z^l = \text{FFN}(\text{LN}(\hat{z}^l)) + \hat{z}^l \qquad (6)$$

where MHSA can be further formulated as:

$$q^{l,m} = z^{l-1}W_q^{l,m}, k^l = z^{l-1}W_k^{l,m}, v^l = z^{l-1}W_v^{l,m} \qquad (7)$$

$$z^{l,m} = \text{softmax}(\frac{q^{l,m}k^{l,mT}}{\sqrt{d}})v^m, m = 1, \cdots, M \qquad (8)$$

$$z^l = \text{concat}(z^{l,1}, \cdots, z^{l,M})W_o^l \qquad (9)$$

where $z^0 = [v, p_1, \cdots, p_N]$, $v$ represents the [CLS] token and $p_1, \cdots, p_N$ are the patch tokens of the query image $x$ with a resolution $H \times W$, and $M$ denotes the number of attention heads.

In image processing, the *Query-Key* retrieval pattern at the final layer can be conceptualized as a type of global average pooling mechanism for capturing visual global descriptions. Concurrently, the *Value* component serves to furnish comprehensive information regarding each position or region within the image. In the current task, our aim is to delve into the interplay between global and local semantic information. Therefore, by adjusting the configuration of the *Value*, whose ensemble forms the output of the final attention mechanism, we can achieve a more nuanced handling of global and local information by the model. Consequently, the model gains the capability to discern the intricate correlation between global and local in a more adaptable manner.

Specifically, for a dense visual input $x_{ij} = x \odot m_{ij}$, where $m_{ij} \in \{0, 1\}^{H \times W}$ represents the mask that is locally active for a kernel around $(i, j)$ and $\odot$ denotes the element-wise product, we can similarly obtain the visual embedding as $f_{l,ij} = f(x \odot m_{ij}) \in \mathbb{R}^d$. In this procedure, the value matrix $v_{L,ij}^l$ for local feature extraction in layer $l$ can be obtained as described in Eq. 7. While the value

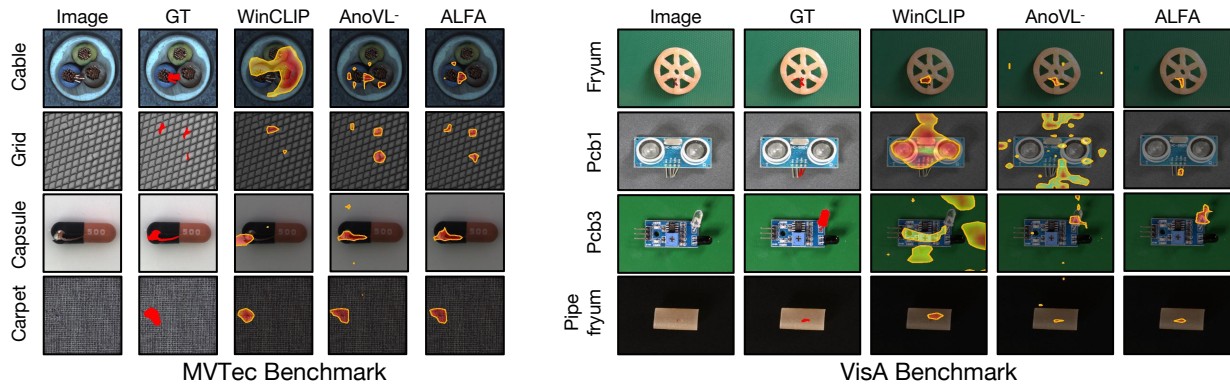

**Figure 5: Qualitative results of zero-shot VAD. Annotated orange regions indicate detected anomalies, showcasing effective localization of ALFA across diverse anomalies (e.g., broken and bent of varying sizes and quantities) within various classes.**

matrix for global feature extraction $f(x)$ in layer $l$ can be similarly represented as $v_G^l$. To this end, we can learn a projection from global to local semantic space by a transformation matrix $W_{T,ij}$ as $W_{T,ij}^l v_G^l = v_{L,ij}^l$.

For the text modality, the global anomaly prompt embedding $\mathcal{F}_{TG} := [f_{TG}^+, f_{TG}^-] \in \mathbb{R}^{2 \times d}$ can be generated by computing embeddings via the text encoder for respective anomaly labels. Next, we project the global anomaly prompt embedding $\mathcal{F}_{TG}$ into the local semantic space as $\mathcal{F}_{TL,ij} = [f_{TL,ij}^+, f_{TL,ij}^-] \in \mathbb{R}^{2 \times d}$ according to each patch token embedding $\mathcal{F}_{IL,ij} = [f_{I,ij}] \in \mathbb{R}^d$ by the transformation matrix $W_{T,ij}$.

After aligning the local embeddings of the anomaly prompt and dense image patch, we calculate the class token-based anomaly score $S_G(x)$ for the image query $x$ and generate an anomaly map using the aligned local embeddings as follows:

$$S_G(x) = \frac{\exp(<f(x), f_{TG}^->)}{\sum_{f_t \in \mathcal{F}_{TG}} \exp(<f(x), f_t>)} \quad (10)$$

$$S_L(x_{ij}) = \frac{\exp(<f_{I,ij}, f_{TL,ij}^->)}{\sum_{f_t \in \mathcal{F}_{TL}} \exp(<f_{I,ij}, f_t>)} \quad (11)$$

Likewise, we can implement multi-scale masked images to generate multi-scale visual embeddings paired with corresponding prompt embeddings. Using these, we calculate multi-scale anomaly maps and average them through harmonic averaging [19] for anomaly localization of a given query. Relying on the premise that an image can be classified as anomalous upon the detection of a single anomalous patch, the anomaly score for the image query is determined by combining the classification score in Eq. 10 with the maximum value of averaged multi-scale anomaly map as follow:

$$S(x) = \frac{1}{2}(S_G(x) + \max_{ij} \widetilde{S_L}(x_{ij})) \quad (12)$$

Our ALFA adeptly accommodates few-shot scenarios by employing a memory bank to store patch-level features from normal samples, illustrated in Figure 2. Anomaly localization is subsequently improved on top of $S_L$ by calculating distances between query patches and their nearest counterparts in the memory bank.

**Table 1: The performance of zero-shot anomaly detection. Bold indicates the best performance.**

| Task | Method | MVTec | | | VisA | | |
|---|---|---|---|---|---|---|---|
| | | AUROC | AUPR | F1-max | AUROC | AUPR | F1-max |
| Image-level | CLIP-AC [35] | 74.1 | 89.5 | 87.8 | 58.2 | 66.4 | 74.0 |
| | WinCLIP [19] | 91.8 | 96.5 | 92.9 | 78.1 | 81.2 | 79.0 |
| | AnoVL⁻ [13] | 91.3 | 96.3 | 92.9 | 76.7 | 79.3 | 78.7 |
| | ALFA | **93.2** | **97.3** | **93.9** | **81.2** | **84.6** | **81.9** |
| Task | Method | pAUROC | PRO | pF1-max | pAUROC | PRO | pF1-max |
| Pixel-level | Trans-MM [5] | 57.5 | 21.9 | 12.1 | 49.4 | 10.2 | 3.1 |
| | MaskCLIP [55] | 63.7 | 40.5 | 18.5 | 60.9 | 27.3 | 7.3 |
| | WinCLIP [19] | 85.1 | 64.6 | 31.7 | 79.6 | 56.8 | 14.8 |
| | AnoVL⁻ [13] | 86.6 | 70.4 | 30.1 | 83.7 | 58.6 | 13.5 |
| | ALFA | **90.6** | **78.9** | **36.6** | **85.9** | **63.8** | **15.9** |

## 5 EXPERIMENTS

In this section, we systematically evaluate ALFA for image-level and pixel-level anomaly detection through quantitative and qualitative analyses on various benchmarks. Ablation studies and explainable VAD results are also presented. Further implementation details and comprehensive experimental results are provided in the Appendix.

### 5.1 Experimental Setup

**Datasets.** Our experiments are based on MVTec [2] and VisA [62] benchmarks, both containing high-resolution images with full pixel-level annotations. MVTec includes data for 10 single objects and 5 textures, while VisA includes data for 12 single or multiple object types. As our framework is entirely training-free, we exclusively utilize the test datasets for evaluation.

**Metrics.** We use Area Under the Receiver Operating Characteristic (AUROC), Area Under the Precision-Recall curve (AUPR), and F1-score at optimal threshold (F1-max) as image-level anomaly detection metrics. Besides, we report pixel-wise AUROC (pAUROC), Per-Region Overlap (PRO) scores, and pixel-wise F1-max (pF1-max) in a similar manner to evaluate anomaly localization.

**Implementation details.** We employ the OpenCLIP implementation [18] and its publicly available pre-trained models. Specifically, we use the LAION-400M [41]-based CLIP with ViT-B/16+ as our foundational model and GPT-3.5 (gpt-3.5-turbo-instruct) for anomaly prompt generation. Further setup details and the prompts list considered in this paper are available in the Appendix.

**Table 2: Component-wise analysis of ALFA on MVTec.**

| RTP adaptation | | | Fine-grained Aligner | Image-level, Pixel-level | | |
|---|---|---|---|---|---|---|
| Template | LLM | $\mathcal{S}_c$ | | (AUROC, pAUROC) | (AUPR, PRO) | (F1-max, pF1-max) |
| × | × | × | × | (34.2, -) | (68.9, -) | (83.5, -) |
| ✓ | × | × | × | (86.6, 79.4) | (92.5, 59.2) | (90.6, 26.8) |
| ✓ | ✓ | × | × | (89.9, 83.6) | (95.2, 62.9) | (92.0, 30.7) |
| ✓ | ✓ | ✓ | × | (92.0, 85.9) | (96.5, 68.8) | (93.0, 32.2) |
| ✓ | ✓ | ✓ | ✓ | **(93.2, 90.6)** | **(97.3, 78.9)** | **(93.9, 36.6)** |

## 5.2 Zero-shot anomaly detection

In Table 1, we compare ALFA with prior arts on MVTec and VisA benchmarks for both image-level and pixel-level zero-shot anomaly detection. Specifically, we compare ALFA with CLIP-AC [35] for image-level anomaly detection, Trans-MM [5] for pixel-level anomaly detection, and WinCLIP [19] and AnoVL [13] for both image-level and pixel-level anomaly detection. For fairness, we use AnoVL⁻ [13] for comparison, representing AnoVL without fine-tuning and data augmentation, while the comparison with the complete AnoVL is presented in Sec. 5.6. More details about the baselines are available in the Appendix A. For both image-level and pixel-level VAD, ALFA demonstrates significant improvements over all baselines across all metrics on both benchmarks. Notably, compared to the runner-up, we achieve a 12.1% enhancement in PRO for pixel-level anomaly detection on MVTec and a 8.9% improvement on VisA. Similarly, for image-level anomaly detection, we outperform the suboptimal method by 1.5% on MVTec and by 4.0% on VisA in terms of AUROC. A detailed breakdown of these gains is presented in Section 5.3 through ablation studies.

**Qualitative results.** In Figure 5, qualitative results for different objects with various anomalies are showcased. In all instances, ALFA yields an anomaly map that exhibits greater concentration on the ground truth compared to previous methods, aligning with the findings from the quantitative results. Subtle, ALFA fares better under various sizes and quantities of anomalies, demonstrating its versatility. More visualizations can be found in the Appendix C.

## 5.3 Ablation Study

**Component-wise analysis.** Ablation studies on key ALFA modules, including RTP adaptation and the fine-grained aligner, demonstrate their significant contributions to overall detection performance, detailed in Table 2. Employing a one-class design using only normal prompts as a baseline in the first row, we emphasize the significance of template-based and LLM-based prompt generator in capturing various anomalous patterns. The contextual scoring mechanism, denoted as $\mathcal{S}_c$ in Table 2, further enhances performance by adaptively managing the anomaly prompts customized for each query image without cross-semantic issue. Furthermore, the fine-grained aligner proves to be a crucial contributor, especially in enhancing pixel-level anomaly detection in zero-shot scenarios.

**Analysis on anomaly prompt.** In Table 3, we demonstrate that ALFA achieves superior detection performance while significantly reducing human efforts in designing prompts. We present the number of prompts per label employed by each method. In general, LVLMs tend to exhibit improved performance with an increase in the number of prompts. However, when cross-semantic ambiguity limits the effectiveness of prompts, increasing their number may not necessarily lead to performance improvement, as evidenced by

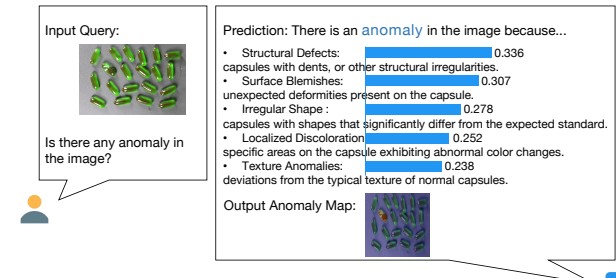

**Figure 6: Interpretable VAD results for capsules in Visa benchmark. The top five descriptors are listed as factors influencing the decision-making.**

**Table 3: Ablation analysis of anomaly prompt on MVTec.**

| #Prompts | Methods | AUROC | AUPR | F1-max |
|---|---|---|---|---|
| 154 (all manual) | WinCLIP [19] | 91.8 | 96.5 | 92.9 |
| 462 (all manual) | AnoVL [13] | 91.3 | 96.3 | 92.9 |
| 146 (only 72 manual) | ALFA with GPT-3 | 92.2 | 96.7 | 93.2 |
| | ALFA with GPT-3.5 | 92.9 | 97.2 | 93.6 |
| | + syntactic consistency | 93.2 | 97.3 | 93.9 |

the results of AnoVL and WinCLIP as shown in Table 3. In ALFA, the range of prompts per label on each class spans from 88 to 216, with an average of 146. Notably, we only design 72 prompts based on the template, a notable 53.2% reduction as compared to over 150 required by baselines, for better detection results, showcasing ALFA's ability to effectively tackle the cross-semantic issue and unlock the full potential of language for zero-shot VAD. We also assess the effect of varying numbers of prompts from template-based and LLM-based generator, with additional results in Appendix B.

We further assess ALFA using GPT-3 (text-davinci-002) and GPT-3.5 (gpt-3.5-turbo-instruct) for automatic prompt generation, observing superior performance with GPT-3.5. To examine the scalability of ALFA, we also conduct tests on various CLIP backbones, with detailed results provided in the Appendix B. Moreover, by augmenting the input query of the GPT-3.5 as "state the description beginning with: An abnormal/normal image of {class}", the resulting prompts are formulated to preserve syntactic consistency to the greatest extent possible, aligning with the text in CLIP pre-training dataset. This augmentation contributes to further improved results.

## 5.4 Interpretability

We present results for explainable anomaly detection in Figure 6, where the bars illustrate the descriptor similarity to the image predicted as an anomaly in the CLIP latent space. Concretely, we condition descriptors on the class name by prompting the language model with the input:

"Q: What are useful descriptions for distinguishing an anomaly {class} in a photo?
A: There are several key descriptions to tell there is an anomaly {class} in a photo:
- "

where "-" is used to generate point-by-point characterizations as descriptors. Figure 6 shows the top five descriptors that emerge from GPT-3.5, encompassing colors, shapes, and object parts for

**Table 4: Image-level performance on few-shot VAD.**

| Setup | Method | MVTec | | | VisA | | |
|---|---|---|---|---|---|---|---|
| | | AUROC | AUPR | F1-max | AUROC | AUPR | F1-max |
| 1-shot | PatchCore [37] | 83.4±3.0 | 92.2±1.5 | 90.5±1.5 | 79.9±2.9 | 82.8±2.3 | 81.7±1.6 |
| | WinCLIP [19] | 93.1±2.0 | 96.5±0.9 | 93.7±1.1 | 83.8±4.0 | 85.1±4.0 | 83.1±1.7 |
| | ALFA | **94.5±1.5** | **97.9±1.4** | **94.9±0.9** | **85.2±2.0** | **87.3±2.1** | **84.9±1.6** |
| 2-shot | PatchCore [37] | 86.3±3.3 | 93.8±1.7 | 92.0±1.5 | 81.6±4.0 | 84.8±3.2 | 82.5±1.8 |
| | WinCLIP [19] | 94.4±1.3 | 97.0±0.7 | 94.4±0.8 | 84.6±2.4 | 85.8±2.7 | 83.0±1.4 |
| | ALFA | **95.9±0.9** | **98.4±0.6** | **95.6±0.6** | **86.4±1.2** | **87.5±1.8** | **85.2±1.4** |
| 4-shot | PatchCore [37] | 88.8±2.6 | 94.5±1.5 | 92.6±1.6 | 85.3±2.1 | 87.5±2.1 | 84.3±1.3 |
| | WinCLIP [19] | 95.2±1.3 | 97.3±0.6 | 94.7±0.8 | 87.3±1.8 | 88.8±1.8 | 84.2±1.6 |
| | ALFA | **96.5±0.6** | **98.9±0.6** | **96.0±0.7** | **88.2±0.9** | **89.4±1.4** | **85.5±1.2** |

**Table 5: Pixel-level performance on few-shot VAD.**

| Setup | Method | MVTec | | | VisA | | |
|---|---|---|---|---|---|---|---|
| | | pAUROC | PRO | pF1-max | pAUROC | PRO | pF1-max |
| 1-shot | PatchCore [37] | 92.0±1.0 | 79.7±2.0 | 50.4±2.1 | 95.4±0.6 | 80.5±2.5 | 38.0±1.9 |
| | WinCLIP [19] | 95.2±0.5 | 87.1±1.2 | 55.9±2.7 | 96.4±0.4 | 85.1±2.1 | 41.3±2.3 |
| | ALFA | **96.8±0.5** | **89.6±1.2** | **57.7±1.6** | **97.2±0.8** | **86.4±1.2** | **42.9±1.9** |
| 2-shot | PatchCore [37] | 93.3±0.6 | 82.3±1.3 | 53.0±1.7 | 96.1±0.5 | 82.6±2.3 | 41.0±3.9 |
| | WinCLIP [19] | 96.0±0.3 | 88.4±0.9 | 58.4±1.7 | 96.8±0.3 | 86.2±1.4 | 43.5±3.3 |
| | ALFA | **97.2±0.4** | **91.0±0.6** | **59.9±1.6** | **97.7±0.8** | **87.2±1.2** | **45.6±2.0** |
| 4-shot | PatchCore [37] | 94.3±0.5 | 84.3±1.6 | 55.0±1.9 | 96.8±0.3 | 84.9±1.4 | 43.9±3.1 |
| | WinCLIP [19] | 96.2±0.3 | 89.0±0.8 | 59.5±1.8 | 97.2±0.2 | 87.6±0.9 | 47.0±3.0 |
| | ALFA | **97.6±0.3** | **91.6±0.6** | **60.3±1.0** | **98.1±0.4** | **89.2±1.2** | **47.9±2.6** |

both class-specific and class-agnostic descriptions. These descriptions enable ALFA to look at cues easily recognizable by humans, enhancing interpretability for decision-making in VAD tasks.

## 5.5 Few-shot Generalization

We expand the capabilities of ALFA to include the few-shot setting, allowing for enhanced performance across scenarios with limited data. We report the mean and standard deviation over 5 random seeds for each measurement in Table 4 and Table 5. We benchmark ALFA against PatchCore [37] and WinCLIP [19]. PatchCore utilizes few-shot images for generating nominal information in its memory bank, and the full-shot version of PatchCore will be discussed in Section 5.6. In this setting, ALFA consistently outperforms all baselines across all metrics, highlighting the efficacy of language prompts and multi-modal alignment for VAD. Moreover, with an increase in the shot number, ALFA exhibits improved performance, emphasizing the synergy between language-driven and reference normal image-based models.

As our anomaly score and anomaly map are dual-composite, we conduct further ablation studies on their distinct components, detailed in Table 6. For image-level VAD, as the number of shots increases, the significance of $S_G$ in anomaly score gradually becomes evident, as it allows the introduction of information from normal images in the memory bank to serve as supervision for VAD. Meanwhile, max $\widetilde{S}_L$ consistently brings further performance improvement based on $S_G$. For pixel-level VAD, we assess the impact of image features generated by masks of various scales, using patches as the unit and a patch size of 16×16 in our foundational model. We also report the average inference time per image across different few-shot settings, evaluated on a server with Xeon(R) Silver 4214R CPU @ 2.40GHz (12 cores), 128G memory, and GeForce RTX 3090. We find that integrating image features at different scales notably improves performance by incorporating local information.

**Table 6: Component-wise analysis of anomaly score and anomaly map on MVTec.**

| Anomaly Score | | #shot (AUROC) | | | |
|---|---|---|---|---|---|
| $S_G$ | max $\widetilde{S}_L$ | 0 | 1 | 2 | 4 |
| ✓ | ✗ | 91.2 | 91.2 | 91.2 | 91.2 |
| ✗ | ✓ | 86.2 | 90.6 | 92.0 | 94.8 |
| ✓ | ✓ | 93.2 | 94.5 | 95.9 | 96.5 |
| Multi-scale | Average | #shots (pAUROC) | | | |
| Mask | Inference Time (s) | 0 | 1 | 2 | 4 |
| [2] | 0.64±0.03 | 88.9 | 93.6 | 95.1 | 95.8 |
| [2, 3] | 1.16±0.06 | 90.6 | 96.8 | 97.2 | 97.6 |
| [2, 3, 4] | 1.92±0.14 | 90.9 | 97.2 | 97.8 | 97.9 |

**Table 7: Comparison of supervised paradigms on MVTec.**

| Methods | Setup | | AUROC | pAUROC |
|---|---|---|---|---|
| | #shots | Training mode | | |
| PaDiM [11] | full-shot | Unsupervised | 84.2 | 89.5 |
| JNLD [54] | full-shot | Unsupervised | 91.3 | 88.6 |
| UniAD [48] | full-shot | Unsupervised | 96.5 | 96.8 |
| AnoVL [13] | 0-shot | Finetuned | 91.3 | 89.8 |
| AnomalyGPT [17] | 0-shot | Finetuned | 97.4 | 93.1 |
| AnomalyCLIP [58] | 0-shot | Finetuned | 91.5 | 91.1 |
| PatchCore [47] | full-shot | Training-free | 99.6 | 98.2 |
| SAA+ [4] | full-shot | Training-free | - | 81.7 |
| MuSc [28] | many-shot (42-176) | Training-free | 97.8 | 97.3 |
| ALFA | 0-shot | Training-free | 93.2 | 90.6 |
| ALFA | 4-shot | Training-free | 96.5 | 97.6 |

However, scaling up the size comes at the cost of increased computational demands, impacting inference speed. Thus, we opt for a scale range of [2, 3] to achieve an optimal trade-off between inference speed and performance.

## 5.6 Comparison on varied supervised paradigms

We benchmark ALFA against prominent unsupervised and finetune-required VAD methods in a unified setting for fairness. Most baselines undergo training or fine-tuning on normal samples encompassing all classes within the dataset. Additionally, we include three full/many-shot training-free methods, PatchCore [37], SAA+ [4] and MuSc [28]. As depicted in Table 7, our zero-shot ALFA is competitive with the baselines that require more information whether in the form of additional normal samples or training. In the 4-shot scenario, ALFA surpasses most baselines, underscoring the complementary roles between language and vision in VAD.

## 6 CONCLUSIONS

In this paper, we present an adaptive LLM-empowered model ALFA that focuses on vision-language synergy for VAD. Capitalizing on the robust zero-shot capabilities of LLMs, the proposed run-time prompt adaptation strategy effectively generates informative prompts by tapping into the vast world knowledge encoded in their billion-scale parameters. This adaptation strategy is complemented by a contextual scoring mechanism, ensuring per-image adaptability while mitigating the cross-semantic ambiguity. Additionally, the introduction of a novel training-free fine-grained aligner further bolsters ALFA, generalizing the alignment projection seamlessly from the global to the local level for precise anomaly localization. Experimental results demonstrate ALFA's superiority over existing zero-shot VAD approaches, providing valuable interpretability.

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
