# OpenReview forum: "Do LLMs Understand Visual Anomalies?  Uncovering LLM's Capabilities in Zero-shot Anomaly Detection"
_acmmm.org/ACMMM/2024/Conference — MM2024 Oral_

### Official Review · Reviewer_tTK7 · 2024-05-25

**Rating:** 5
**Confidence:** 4

**Summary:**

The authors employ the Large vision-language models to deal with the problem of zero-shot VAD. They propose a run-time prompt adaptation strategy, which first generates informative anomaly prompts to leverage the capabilities of a large language model (LLM). This strategy is enhanced by a contextual scoring mechanism for per-image anomaly prompt adaptation and cross-semantic ambiguity mitigation. They further introduce a novel fine-grained aligner to fuse local pixel-level semantics for precise anomaly localization, by projecting the image-text alignment from global to local semantic spaces. Extensive evaluations on the challenging MVTec and VisA datasets confirm ALFA's effectiveness in harnessing the language potential for zero-shot VAD, achieving significant PRO improvements of 12.1% on MVTec AD and 8.9% on VisA compared to state-of-the-art zero-shot VAD approaches.

**Strengths:**

The method is simple and easy to understand, the motivation is clear, the logical structure is reasonable, and it leverages the advantages of popular large vision-language models. The problem addressed is highly practical.

**Limitations:**

1. What are the shortcomings of your method?
2. Tables 4 and 5 need to include comparisons with more recent methods.
3. More experimental analysis is needed to demonstrate the effectiveness of the method.

**Suitability:**

3

---

### Official Review · Reviewer_qhy3 · 2024-05-26

**Rating:** 5
**Confidence:** 3

**Summary:**

This paper addresses zero-shot image-level anomaly detection (AD) with LLM's anomaly description generation capability. Motivated by the weakness of class-conditional anomaly description generation when image context is unavailable, this paper proposes two components, i.e., run-time anomaly prompt scoring mechanism and fine-grained aligner, which reduces the ambiguity of anomaly text description. Experiments verify the superiority of the proposed method for zero-shot AD.

**Strengths:**

Overall, the paper is of good quality.
The main strengths are that:
(1) The motivation to address the ambiguity of anomaly description when image context is not used is clear.
(2) Accordingly, this paper proposes several components to adapt the high-level representation/knowledge from frozen LLM to the zero-shot fine-grained AD task.
(3) The proposed method is verified to be effective.

**Limitations:**

I have only a few minor comments on the experimental part.
(1) Although this method tries to make the prompt generation process more automatic, the proposed method still requires a large number of anomaly descriptions through manual designs. How much does the accuracy and number of anomaly descriptions affect the performance?
(2) It would be helpful to provide the computational cost, as some people in the AD community would emphasise the practical applicability of AD methods.
(3) It would be interesting to test the upper bound of the proposed method with much more powerful LLMs such as GPT-4v. That is, if the LLM is much more powerful, would this resolve/reduce the ambiguity of the anomaly description?

**Suitability:**

2

---

### Official Review · Reviewer_9zDq · 2024-05-27

**Rating:** 4
**Confidence:** 3

**Summary:**

The paper proposes a training-free zero-shot visual anomaly detection model called ALFA that aims to leverage the capabilities of language models to improve zero-shot anomaly detection performance. This strategy utilizes a contextual scoring mechanism for adaptive prompt adaptation and semantic disambiguation for each image. Additionally, a novel fine-grained aligner is introduced, which projects the image-text alignment from global semantic space to local semantic space, enabling precise anomaly localization at the pixel level.

**Strengths:**

1. The article proposes the ALFA model as a training-free zero-shot visual anomaly detection method. ALFA generates adaptive anomaly prompts and achieves precise pixel-level anomaly localization. It leverages a large language model to generate informative anomaly prompts and adapts them on a per-image basis using a contextual scoring mechanism.
2.The ALFA model introduces a fine-grained aligner that generalizes the projection from global to local semantic space, enabling global and local zero-shot anomaly detection without requiring additional data or fine-tuning.
3.The proposed approach is evaluated comprehensively on MVTec and VisA benchmarks, demonstrating the effectiveness of the ALFA model in zero-shot anomaly detection. Furthermore, the ALFA model can be easily extended to few-shot settings, achieving state-of-the-art results that are on par with or even surpass full-shot and fine-tuning-based methods.

**Limitations:**

（1）Word Choice and Grammar Corrections:
In line 88, the phrase “and etc” should be revised to simply “etc.” as “etc.” already implies the continuation of a list.
The spelling of “run-time” in the description of Figure 2 on line 250 appears to be incorrect. It should be unified to “runtime” throughout the document.
There seems to be a grammatical inconsistency in line 723 with the phrase “in terms of AUROC.” It would be more accurate to revise it to “in AUROC.”
In line 765, the term ‘VisA’ appears to be misspelled as ‘Visa’. It is crucial to ensure the correct capitalization is used throughout the document for consistency and accuracy.”
（2）Insufficient Explanation of Model Utilization:
The paper could benefit from a more detailed explanation of how the model leverages a memory bank to improve defect localization in scenarios with few-shot.
（3）Questions Regarding Model Generalizability:
Since the evaluation dataset employed is industrial, it raises the question of whether the model is solely capable of detecting defects in industrial settings or if it can be generalized to detect defects in other sectors as well.
（4）Model Limitations and Future Work:
It would be constructive for the manuscript to discuss any limitations of the model and suggest avenues for future research and development.

**Suitability:**

2

---

### Official Review · Reviewer_odX9 · 2024-06-09

**Rating:** 4
**Confidence:** 4

**Summary:**

This manuscript proposes a novel adaptive LLM-empowered model ALFA for zero-shot anomaly detection, which focuses on vision-language synergy. Specifically, this method proposes a prompt adaptation strategy to mitigate the cross-semantic ambiguity by a contextual scoring mechanism. Furthermore, the method introduces a novel fine-grained aligner to fuse local pixel-level semantics for precise anomaly location. The experiment results demonstrate the superiority of the proposed framework to some extent.

**Strengths:**

1. This paper designs modules from the perspective of cross-modal fine-grained alignment, which is innovative in the field of zero-shot anomaly detection.
2. The proposed method is straightforward and has theoretical feasibility. Meanwhile, this paper conducts experiments from multiple perspectives to prove the effectiveness of the proposed method compared with other SOTA methods.
3. Theoretically, all VLM-based anomaly detection models can be fine-grained aligned based on the idea of this paper, which has a relatively high application value.

**Limitations:**

1. The component-wise ablation experiments do not seem to be complete and are not performed for each case individually. It would be better to complete the ablation experiments.
2. The visualization results of different methods are compared in Fig. 5, however, the precise localization ability of the proposed method is not reflected. It is desirable to reflect the performance difference in anomaly localization before and after using the aligner through some visualization results.

**Suitability:**

3

---

### Meta-Review · Area_Chair_xYJg · 2024-06-30

**Recommendation:** Accept (Oral)
**Confidence:** 5

**Metareview:**

The paper initially received positive indications (2BA, 2WA), which were later raised after the rebuttal phase (3WA final ratings + 1 preliminary BA not uploading the final rating). Some concerns raised during the initial reviews have been cleared during the rebuttal, so the paper can be accepted. Considering the ratings of the paper, the quality of the reviews, and the quality of the paper perceived by the AC, the paper can also be accepted as oral.